# Influence of Venoarterial Extracorporeal Membrane Oxygenation Integrated Hemoadsorption on the Early Reversal of Multiorgan and Microcirculatory Dysfunction and Outcome of Refractory Cardiogenic Shock

**DOI:** 10.3390/jcm11216517

**Published:** 2022-11-02

**Authors:** Adam Soltesz, Zsofia Anna Molnar, Zsofia Szakal-Toth, Eszter Tamaska, Hajna Katona, Szabolcs Fabry, Gergely Csikos, Viktor Berzsenyi, Csilla Tamas, Istvan Ferenc Edes, Janos Gal, Bela Merkely, Endre Nemeth

**Affiliations:** 1Heart and Vascular Center, Semmelweis University, H-1122 Budapest, Hungary; 2Department of Anesthesiology and Intensive Therapy, Semmelweis University, H-1085 Budapest, Hungary

**Keywords:** refractory cardiogenic shock, venoarterial extracorporeal membrane oxygenation, hemoadsorption, cytosorb, sequential organ failure assessment score

## Abstract

Background: The purpose of this investigation was to evaluate the impact of venoarterial extracorporeal membrane oxygenation (VA–ECMO) integrated hemoadsorption on the reversal of multiorgan and microcirculatory dysfunction, and early mortality of refractory cardiogenic shock patients. Methods: Propensity score–matched cohort study of 29 pairs of patients. Subjects received either VA–ECMO supplemented with hemoadsorption or standard VA–ECMO management. Results: There was a lower mean sequential organ failure assessment score (*p* = 0.04), lactate concentration (*p* = 0.015), P_(v–a)_CO_2_ gap (*p* < 0.001), vasoactive inotropic score (*p* = 0.007), and reduced delta C–reactive protein level (*p* = 0.005) in the hemoadsorption compared to control groups after 72 h. In–hospital mortality was similar to the predictions in the control group (62.1%) and was much lower than the predicted value in the hemoadsorption group (44.8%). There were less ECMO-associated bleeding complications in the hemoadsorption group compared to controls (*p* = 0.049). Overall, 90-day survival was better in the hemoadsorption group than in controls without statistical significance. Conclusion: VA–ECMO integrated hemoadsorption treatment was associated with accelerated recovery of multiorgan and microcirculatory dysfunction, mitigated inflammatory response, less bleeding complications, and lower risk for early mortality in comparison with controls.

## 1. Introduction

Cardiogenic shock (CS) is a complex acute hemodynamic syndrome with an early mortality rate as high as 40–90% depending on its etiology and resistance to conventional pharmacotherapy [1,2,3,4]. The primary cause of death is multiorgan dysfunction, which is triggered by the persistent low cardiac output state and consecutive organ and tissue hypoperfusion [5,6]. This process is further aggravated by microcirculatory dysfunction linked to dysregulated activation of proinflammatory cytokines, complements, and excessive release of nitric oxide [6,7,8]. Over the last two decades, temporary mechanical circulatory support (MCS) technology has become a pivotal tool in the acute care of refractory cardiogenic shock aiming to facilitate the rapid restoration of macro- and micro-hemodynamics and prevent the emerging multiorgan dysfunction from progressing to an irreversible phase [6,9]. Despite the application of MCS in the complex therapy of refractory CS, the survival to discharge remains 45% according to 2021 data from the Extracorporeal Life Support Registry Report summary [10].

Venoarterial extracorporeal membrane oxygenation (VA–ECMO) is among the MCS modalities most frequently used in the acute care of refractory CS [2,9]. While VA–ECMO is effective in supporting macrocirculatory hemodynamics to rapidly normalize, the same benefit on CS-associated microcirculatory dysfunction and impaired tissue oxygen delivery remains controversial [10,11,12]. Additionally, recent investigations have demonstrated that ineffective recruitment of functional capillary density during the early phase of VA–ECMO support can be associated with worse outcomes in refractory CS [11,12,13]. The mechanisms that contribute to rapid recovery of persisting microcirculatory dysfunction after VA–ECMO initiation have not yet been discovered [10]. However, adverse interactions of pathophysiological factors such as elevated plasma free hemoglobin (fHb) levels, depleted soluble guanylyl cyclase activity, and concomitant increased platelet aggregation and amplified dysregulated proinflammatory response linked to the application of extracorporeal MCS can negatively influence the normalization of the microcirculatory function [10].

Cytokine hemoadsorption is an extracorporeal blood purification technology, which is effective in the elimination of pro- and anti-inflammatory cytokines, chemokines, bilirubin, myoglobin, plasma fHb, and several pharmacological agents [14,15,16]. Some recent case series and observational studies have reported on a mitigated inflammatory response, improved hemodynamics, metabolic stability, and improved outcome trends associated with hemoadsorption when it has been tested in patients who require VA–ECMO support [17]. However, a most recent retrospective, propensity score matched analysis did not confirm any advantage of VA–ECMO integrated hemoadsorption on outcome in patients who received extracorporeal cardiopulmonary resuscitation (ECPR) [18]. Indeed, the availability of high-quality clinical investigations that clarify the clinical utility of hemoadsorption treatment during VA–ECMO support is scarce; an important field therefore remains vital [17].

The aim of this retrospective study was therefore to analyze the clinical impact of VA–ECMO integrated hemoadsorption in terms of early reversal of multiorgan- and microcirculatory dysfunction, and short-term clinical outcomes in patients undergoing VA–ECMO support for refractory CS, using propensity score matching.

## 2. Materials and Methods

### 2.1. Ethical Statement

This study was approved by the Regional and Institutional Committee of Science and Research Ethics and was conducted in accordance with the Declaration of Helsinki. The requirement for written informed consent was waived by the ethics committee because of the retrospective observational design of this study (Number of the ethical approval: 72/2022; Date of approval: 11 April 2022).

### 2.2. Patients and Data Collection

This study analyzed retrospectively collected clinical data of adult patients supported with VA–ECMO due to refractory cardiogenic shock between 1 January 2012 and 31 December 2020. Clinical characteristics, follow-up, and outcome data, along with acute physiology and chronic health evaluation II (APACHE II) [19], sequential organ failure assessment (SOFA) [20], and survival after venoarterial extracorporeal membrane oxygenation (SAVE) scores [21], arterial and venous blood gas variables obtained from the digital databases of the Cardiovascular Critical Care Unit and the Hospital Healthcare System, as well as data from individual treatment charts (intensive care observational charts) were collated. Patients who died within 72 h or did not develop vasoplegia syndrome were excluded from the extended analyses. Over the screened period there were no relevant changes in the indication criteria for VA–ECMO support and all patients received standardized intensive care of VA–ECMO management.

### 2.3. Venoarterial Extracorporeal Membrane Oxygenation Management

VA–ECMO support was provided using the Medos Deltastream System (Medos Medizintechnik AG, Stolberg, Germany). Patients received peripheral (i.e., femoral) or central cannulation for the VA–ECMO circuit according to the etiology of the refractory CS. Peripheral VA–ECMO circuit was extended by a femoral distal perfusion catheter in all cases. Initial VA–ECMO support was adjusted to achieve blood flow rates of 3.0–4.0 L/min, which was supplemented by an additional 500 mL/min if hemoadsorption treatment was also introduced. Administration of pharmacologic circulatory support (i.e., vasopressor and/or inotropic agents) and intravenous fluid therapy was set to hemodynamic goals of mean arterial pressure ≥ 70 mmHg and aortic valve opening in each cardiac cycle. Systemic anticoagulation was maintained with unfractionated heparin and antithrombin III (AT–III) targeting an activated partial thromboplastin time ratio of 2.0–2.5 and serum AT–III activity ≥ 70%. After the completion of 3–5 days of optimized VA–ECMO support, all patients received standardized stepwise VA–ECMO weaning (200–300 rate/minute decrease 12–24 hourly up to 2.0 L/minute flow support depending on cardiac performance and hemodynamic response) for the subsequent days in accordance with the institutional protocol. Patients were candidates for VA–ECMO explanation after successful weaning, including a 24-h period on low flow support (2.0 L/minute). In case of persistent MCS dependence, patients were converted to a mid-term MCS device.

### 2.4. Hemoadsorption Treatment

The first continuous VA–ECMO integrated hemoadsorption treatment was performed at our institution in 2017 and previously published in detail [22]. Patients were candidates for hemoadsorption treatment if they presented with vasoplegia syndrome, defined as a norepinephrine requirement ≥ 0.3 μg/kg/min and the need for arginin–vasopressin at any dose, 4–6 h after VA–ECMO initiation, despite combined hemodynamic resuscitation. However, the application of hemoadsorption was at the discretion of the treating physician due to lack of clinical recommendation in this field. Hemoadsorption was performed using CytoSorb^TM^ 300 mL cartridge (Cytosorbents^TM^, Monmouth Junction, NJ, USA) incorporated into the VA–ECMO circuit for a 72-h continuous treatment in total.

### 2.5. Outcome Parameters

The primary outcomes of this study were the change in SOFA score after 72 h of VA–ECMO run and in–hospital mortality. Secondary outcome parameters were defined as early metabolic stability, change in microcirculatory function described by the P_(v–a)_CO_2_ gap (P_(v–a)_CO_2_ gap = P_v_CO_2_ − P_a_CO_2_), inflammatory activity characterized by C–reactive protein (CRP) and white blood cell (WBC) count, hemodynamic stability described by vasoactive inotropic score (VIS = dopamine dose (μg/kg/min) + dobutamine dose (μg/kg/min) + 100 × adrenaline dose (μg/kg/min) + 10 × phosphodiesterase inhibitor dose (μg/kg/min) + 100 × noradrenaline dose (μg/kg/min) + 10,000 × vasopressin dose (U/kg/min) [23] based on the actual doses of each agent) in a time frame of the first 72-h VA–ECMO support, major complications associated with refractory CS and VA–ECMO support, intensive care unit and hospital stay, and 90-day survival. Stages of acute kidney injury were classified applying the Kidney Disease Improving Global Outcomes creatinine-based definition criteria for the first 72-h time frame of the VA–ECMO support [24].

### 2.6. Statistical Analysis

Descriptive statistics of data were presented as mean ± standard deviation, while categorical variables were displayed as the number of patients and frequency. We performed a 1:1 match, nearest neighbor method propensity score matching (PSM) with a caliper width of 0.2 [25] using the logistic regression estimation algorithm with adjusted covariates from the APACHE II and SOFA scores, average ECMO flow, and postcardiotomy etiology of refractory CS. The comparative analyses of continuous and categorical variables, including within-subjects changes in the matched cohort, were accomplished with the paired *t*–test and McNemar test, where appropriate. We completed 90-day follow-up for all included patients and estimated the 90-day survival for the two matched groups using the Kaplan–Meier method. The equality testing of survival curves was performed with the stratified log–rank test involving the quintiles of the estimated propensity scores as strata [26,27]. Statistical significance was defined at the 0.05 level in all tests. Analyses were performed with IBM SPSS Statistics for Windows, version 27.0 (IBM Corp., Armonk, NY, USA) and R–statistics for Windows, version 3.6.0 (R Foundation for Statistical Computing, Vienna, Austria).

## 3. Results

### 3.1. Clinical Characteristics

Overall, 268 patients were treated with refractory CS and VA–ECMO support in the investigated period at our institution. After the exclusions, the PSM procedure involving 150 patients resulted in 29 matched pairs (Figure 1). The absolute values of standardized mean differences were found to be less than 0.225 for all adjusted covariates. APACHE II and SOFA scores achieved balance by PSM, which indicated similar risks for early mortality in both groups prior to VA–ECMO implantation. The univariate analyses of the baseline parameters did not reveal relevant differences between the two groups in terms of patient characteristics (Table 1). However, the peripheral VA–ECMO support was less frequent in patients of the hemoadsorption group than the control group. The patient selection process and the clinical characteristics in the unmatched and matched cohorts are summarized in Figure 1 and Table 1, respectively.

### 3.2. Primary Outcomes

Subjects from the hemoadsorption group experienced a significant reduction in the follow-up 72-h SOFA score from 12.1 ± 2.8 to 10.1 ± 3.3 (*p* < 0.001), with no difference detected in the control group (12.2 ± 1.8 versus 12.1 ± 3.7, *p* = 0.815, respectively; Figure 2). Additionally, the 72-h SOFA score was also significantly lower in the hemoadsorption than in the control group (Table 2). We registered a higher frequency of in-hospital mortality in the control compared to the hemoadsorption groups (62.1% vs. 44.8%, respectively), however, this difference was not statistically significant (Table 2). Interestingly, the observed in-hospital mortality was also lower than the mean predicted value calculated according to the APACHE II and SOFA scores prior to VA–ECMO initiation in patients from the hemoadsorption group (44.8% vs. 63.1% and 73.2%, respectively), while there were no relevant differences in the controls (62.1% vs. 59.3% and 74.4%, respectively, Figure 3).

### 3.3. Secondary Outcomes

The mean lactate level decreased significantly in both the control and hemoadsorption groups 72 h after VA–ECMO initiation (2.11 vs. 6.90 mmol/L, *p* < 0.001, 1.57 vs. 6.56 mmol/L, *p* < 0.001, respectively). Nevertheless, the mean lactate was found to be in the normal range and significantly lower in the hemoadsorption than the control group, which persisted outside the lactate upper limit in the latter group at the 72-h follow-up time point (Table 2). Similarly, the P_(v–a)_CO_2_ gap declined significantly and normalized after 72 h of VA–ECMO support in subjects from the hemoadsorption group (4.47 vs. 8.47 mmHg, *p* < 0.001), while the P_(v–a)_CO_2_ gap remained elevated and in the pre–ECMO range in the controls (8.13 vs. 9.19 mmHg, *p* = 0.109). We observed a significant reduction in VIS in the two groups during the first 72-h time frame of VA–ECMO run (control group: 79.2 ± 51.0 vs. 35.2 ± 36.1 points, *p* < 0.001 and hemoadsorption group: 90.0 ± 61.7 vs. 13.8 ± 19.5 points, *p* < 0.001). Additionally, the VIS of the hemoadsorption group was significantly lower comparing to that of the control group (*p* = 0.007, Table 2). The mean CRP showed an increase up to similar ranges in both the control and hemoadsorption groups (140.05 mg/L vs. 116.69 mg/L, *p* = 0.159, respectively, Table 2) after 72 h of VA–ECMO support. However, the magnitude of the CRP change (delta CRP) was significantly smaller in the hemoadsorption than in the control group (50.13 ± 85.29 mg/L vs. 108.47 ± 87.20 mg/L, *p* = 0.005, respectively, Figure 4). The length of mechanical ventilation, intensive care unit, and hospital stays were comparable in the two groups. Early major complications, registered for the first 72 h of the VA–ECMO support, did not show relevant differences, except for clinically relevant bleeding related to the VA–ECMO application. While this complication had a significantly lower frequency in the hemoadsorption versus control group, the rate of reoperation for bleeding was similar in both groups (Table 2). Detailed analyses of the primary and secondary outcome parameters from the hemoadsorption and control groups are shown in Table 2. Analysis of cumulative 90-day survival did not reveal a statistically significant difference between the groups; however, there was a trend towards improved mortality in the hemoadsorption group compared to the control group for the complete observational period (Figure 5).

## 4. Discussion

In this retrospective, propensity score-matched cohort study we aimed to analyze the clinical significance of VA–ECMO integrated hemoadsorption in terms of early reversal of multiorgan and microcirculatory dysfunction, and short- and long-term outcomes of critically ill patients treated with refractory CS compared to standard VA–ECMO management. Our study demonstrated that patients who received a 72-h period of hemoadsorption treatment showed a significant reduction in SOFA score, faster normalization of macrohemodynamics, metabolic state and P_(v–a)_CO_2_ gap, and lower risk for early mortality than patients in the control group. VA–ECMO integrated hemoadsorption treatment was associated with reduced delta CRP and less bleeding complications compared with the controls.

Multiorgan dysfunction is a dominant contributing factor of in-hospital mortality risk associated with refractory CS [6]. The SOFA score is a widely employed composite assessment tool in critical care to classify and monitor multiorgan dysfunction over time [29]. SOFA score assessed prior to VA–ECMO initiation has been found to have good predictive value for in-hospital mortality in earlier clinical investigations of patients undergoing VA–ECMO support [30,31,32]. Similarly, a decreasing SOFA score at day 3 of VA–ECMO support has been associated with better hospital survival in the same clinical scenario, demonstrating the link between the early improvement of organ function and outcome [31,33]. In our study, we observed significantly reduced mean SOFA scores in the hemoadsorption group after 72 h of VA–ECMO start compared to the initial value (Figure 2). Despite the identical combined mechanical and pharmacological circulatory support applied in the control group the mean 72-h SOFA score persisted in the pre–ECMO range in these subjects. Only very few clinical studies and case series have previously examined the significance of ECMO integrated hemoadsorption on patient outcome–among them, two comparative investigations involving VA–ECMO patients [17,18,34]. Of these two studies only the randomized controlled trial (RCT) published by Supady et al. used longitudinal SOFA score follow-up [34]. They did not find any significant differences in either the longitudinal change or the 72-h values of the SOFA scores [34]. However, 54.5% of patients in the cytokine adsorption group, and 73.7% of patients in the control group compared to baseline survived the 72-h timepoint in their study, which restricts the interpretation of SOFA score change in the study groups [34]. In our analysis, we excluded patients who died on VA–ECMO within 72 h to mitigate patient selection bias in the advanced analyses, which resulted in the complete comparison of groups in terms of SOFA score change. Interestingly, in a recent RCT including patients with severe COVID–19 pneumonia requiring venovenous ECMO, a marked reduction in SOFA score was seen in the cytokine adsorption group versus controls with a time frame of 72 h, despite the lower range of initial SOFA scores registered in the groups [35]. These results are in line with the findings of our study supporting the assumption that VA–ECMO integrated hemoadsorption can contribute to accelerated the reversal of multiorgan dysfunction induced by refractory CS.

Our analysis confirmed significant reduction of VIS in both groups over the first 72 h of VA–ECMO support demonstrating an obvious stabilization of the macrohemodynamics. This change of VIS was more robust in the hemoadsorption group than the control group (Table 2). However, the restoration of macrohemodynamics during adequate VA–ECMO support does not result in instant and simultaneous improvement in microcirculatory dysfunction [10]. Moreover, prolonged impairment of microhemodynamics and tissue oxygen delivery can be an independent factor of unfavorable outcome in patients receiving VA–ECMO support [10,11,12]. Indeed, ECMO associated pathomechanisms involving plasma fHb and dysregulated inflammatory response linked processes can amplify microcirculatory dysfunction and delay its normalization [10]. In this context, the integration of the hemoadsorption treatment into a VA–ECMO system early on in the clinical course can theoretically control the adverse microcirculatory effects of the aforementioned pathophysiological interferences [36]. As a surrogate marker of hemodynamic coherence and microcirculatory function, the P_(v–a)_CO_2_ gap was monitored in VA–ECMO patients in a recent retrospective cohort study [37]. They found that an elevated P_(v–a)_CO_2_ gap measured in the initial course of VA–ECMO support was associated with poor outcome [37]. Our data show a significantly lower and normalized 72-h P_(v–a)_CO_2_ gap and lactate level in patients from the hemoadsorption group than controls (Table 2). Both parameters suggest early reversal of microcirculatory dysfunction and impaired tissue oxygen delivery in the hemoadsorption group, which was delayed in the controls according to their persistently elevated mean P_(v–a)_CO_2_ gap and lactate level registered at 72 h. Considering the results of our investigation it can be supposed that a 72-h VA–ECMO integrated hemoadsorption treatment can contribute to the rapid reversal of macro- and microcirculatory dysfunction and restoration of hemodynamic coherence, resulting in improved organ function.

Hemoadsorption as an adjunctive treatment can be considered in clinical conditions in which hyperinflammation and a consecutive impaired hemodynamics are present [36]. Previous case reports and case series demonstrated marked reductions in CRP, procalcitonin, and interleukin–6 (IL–6) related to hemoadsorption treatment combined with VA–ECMO support [22,38,39]. Nevertheless, most recent PSM- and RCT-based analyses of ECPR patients found comparable CRP and IL–6 levels in both the cytokine adsorption and control group after 72 h of VA–ECMO run [18,34]. The results of our study are different from findings of the latter investigations. While the mean 72–hour levels of CRP were in a similar range in the groups, the magnitude of delta CRP was significantly smaller in the patients from the hemoadsorption than the control group (Figure 4), suggesting a mitigated inflammatory response. The possible explanation for this discrepancy can be the divergent patient selections used in the investigations. Unlike the former studies that analyzed ECPR patients, we investigated unselected refractory cardiogenic shock patients that received VA–ECMO support, with 62.1% of postcardiotomy cases in each group. Additionally, the more significant immune system priming along with higher mortality rate within 72 h of patients presented in the cytokine versus control group in the CYTER study assume relevant differences in terms of the severity of initial multiorgan dysfunction as well as the intensity of the inflammatory response between the analyzed groups, which can influence the interpretation of the detected levels of the inflammatory markers [34]. Furthermore, the significantly smaller mean delta CRP measured in the hemoadsorption group in our study is in line with the findings of the reduced mean SOFA score, lactate level, and P_(v–a)_CO_2_ gap at the 72-h follow-up point compared to controls indicating the role of the inflammatory control provided by the continuous hemoadsorption in the early reversal of the refractory CS associated multiorgan dysfunction.

This study analyzed cohorts of patients with various etiologies for refractory CS. However, both the unmatched and matched cohorts presented comparable frequencies of the typical CS etiologies with previously reported data (Table 1) [40]. Due to the between-group comparison of clinical characteristics including the major CS etiologies, APACHE II and SOFA composite scores did not reveal any differences in the PSM cohort; we presumed identical risks for complications and early mortality. The frequency of in-hospital mortality was 62.1% in the control group, which is congruent with the mean predicted values of the pre–ECMO APACHE II and SOFA scores (Figure 3). Additionally, the observed in-hospital mortality of the control group is in line with recently published data of non-selected and post-cardiotomy VA–ECMO patients, demonstrating an in-hospital survival rate between 34.4% and 43.4% [31,41,42,43,44]. On the other hand, we registered lower in-hospital mortality (44.8%) and better 90-day survival in the hemoadsorption group than in controls in our study. Although these marked differences in the mortality and survival outcome did not reach statistical significance, they are indicative of an early mortality risk reduction to ~50% that might be a result of the improvement in microcirculatory and multiorgan dysfunction linked to the VA–ECMO integrated hemoadsorption treatment. Among the major complications, the observed number of ECMO-associated bleeds showed a significant difference between the two groups. The definition of bleeding complications regarding VA–ECMO support shows large diversity in the publications, which makes for limited comparison possible between the observed and registry data [40]. Furthermore, none of the published investigations have examined the frequency of bleeding related to VA–ECMO integrated hemoadsorption treatment to date. In our study, we have defined an ECMO-associated bleeding complication as a clinically relevant event requiring conservative therapy (i.e., blood products and factor concentrates) and/or surgical therapy. Considering the differences in the total number of bleeding events and reoperation rates, our data suggest that the dominant component of the between-group discrepancy is the minor bleeding complication, because the reoperation rates were similar in the groups (Table 2). This result from our study raises the possibility that the more frequent instability of the hemostatic system during the early phase of the VA–ECMO support in the control versus hemoadsorption group could be a part of the persistent multiorgan dysfunction presented by the significantly higher mean 72-h SOFA score in the control group. On the other hand, the controlled ECMO circuit-induced inflammatory processes achieved by the continuous hemoadsorption treatment could also contribute to stabilize hemostatic system indirectly gaining more restraint in terms of diffuse bleeding at the surgical sites.

## 5. Limitations

The present study has some limitations. Due to the retrospective design applied in this investigation, we performed the PSM modelling approach to minimize the characteristic discrepancy linked bias. Nevertheless, some hidden confounders may be present. The therapeutic utilization of VA–ECMO integrated hemoadsorption was not strictly protocolized in the study period, and the clinical decision whether to start hemoadsorption treatment or not was at the discretion of the treating physician. Considering these limitations and the sample size of the analyzed cohort in part restricts the interpretation of our results.

## 6. Conclusions

This propensity score-matched cohort study demonstrates that patients who received a 72-h length VA–ECMO integrated hemoadsorption treatment realized significant reductions in their SOFA score, faster normalization of macrohemodynamics, metabolic state, and P_(v–a)_CO_2_ gap over the same time frame than subjects in the control group, suggesting accelerated recovery of CS-associated multiorgan and microcirculatory dysfunction. VA–ECMO integrated hemoadsorption treatment was associated with mitigated inflammatory response, less bleeding complications, and lower risk for early mortality predicted by the APACHE II and SOFA composite scores in comparison with controls. However, further prospective studies are warranted in this field to clarify the potential benefits of VA–ECMO integrated hemoadsorption in patients treated with refractory CS.

## Figures and Tables

**Figure 1 jcm-11-06517-f001:**
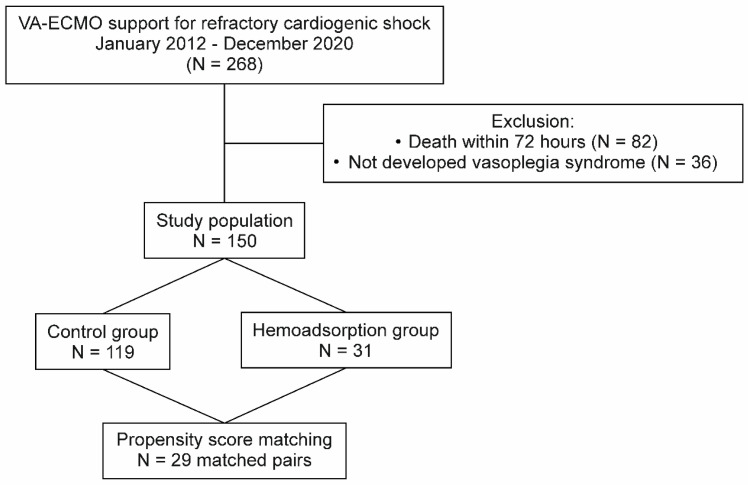
Patient selection flowchart. VA–ECMO: venoarterial extracorporeal membrane oxygenation.

**Figure 2 jcm-11-06517-f002:**
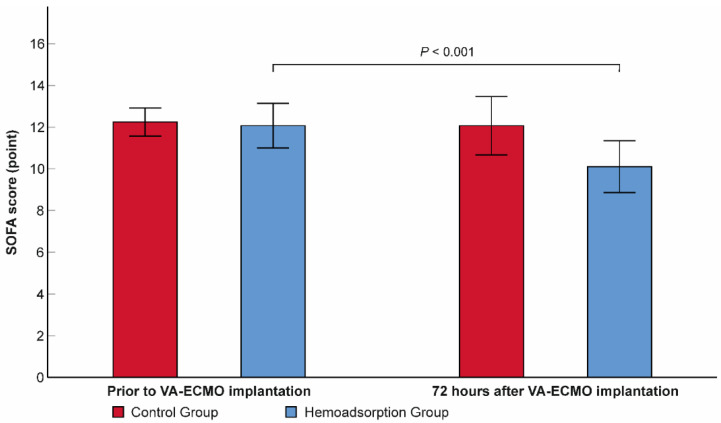
Within-subjects change in sequential organ failure assessment score over the first 72 h of venoarterial extracorporeal membrane oxygenation support. N = 58. Data are presented as means. Error bars show 95% confidence intervals. SOFA: sequential organ failure assessment; VA–ECMO: venoarterial extracorporeal membrane oxygenation.

**Figure 3 jcm-11-06517-f003:**
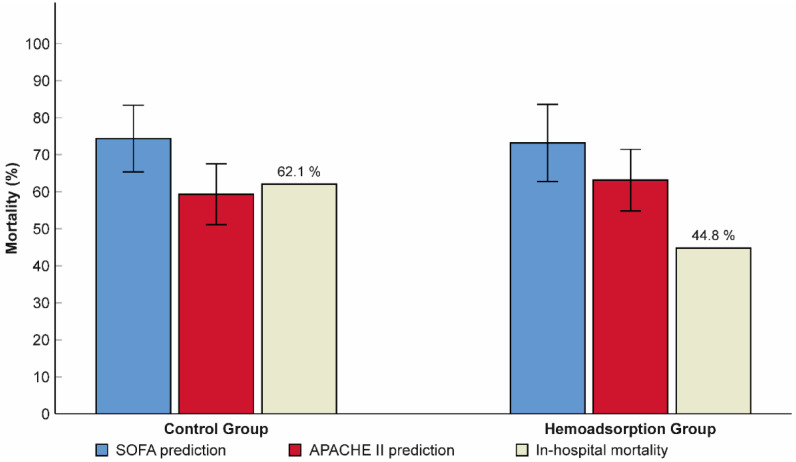
Relationship between predicted and observed in-hospital mortality rates in patients from the hemoadsorption and control groups. N = 58. Data are presented as means and frequency (%). Error bars show 95% confidence intervals. SOFA: sequential organ failure assessment; APACHE II: acute physiology and chronic health evaluation II.

**Figure 4 jcm-11-06517-f004:**
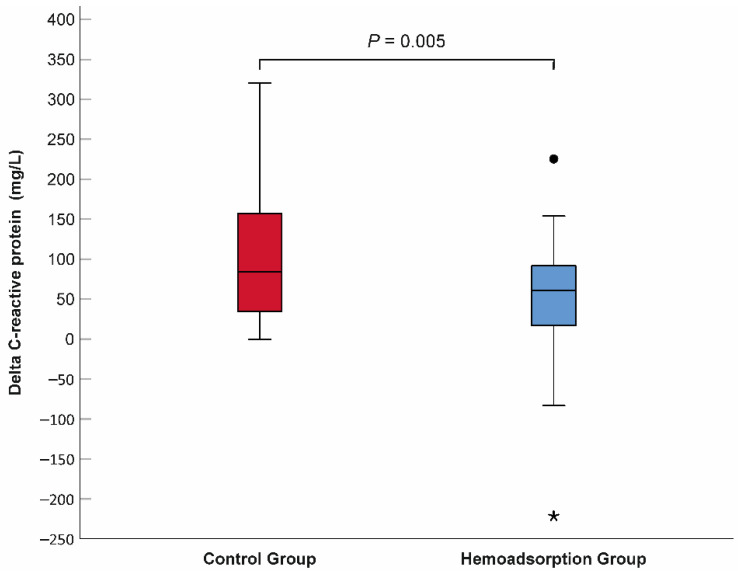
Comparison of delta C-reactive protein between the hemoadsorption and control groups. N = 58. Delta C-reactive protein = 72-h CRP–baseline CRP. Filled circle indicates outlier, while asterisk represents extreme value.

**Figure 5 jcm-11-06517-f005:**
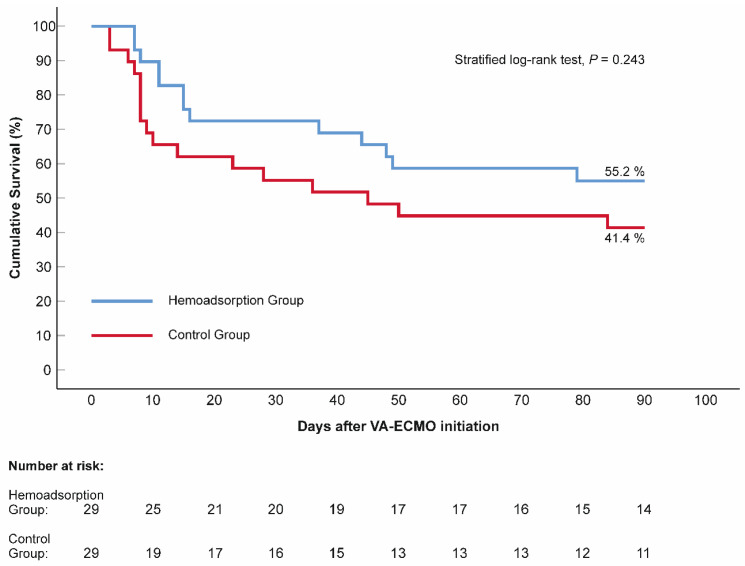
Kaplan–Meier estimates of cumulative 90-day survival, according to the applied venoarterial extracorporeal membrane oxygenation management. N = 58. The blue line represents the hemoadsorption group, while the red line illustrates the control group. *p* value (stratified log–rank test) indicates the difference in survival. VA–ECMO: venoarterial extracorporeal membrane oxygenation.

**Table 1 jcm-11-06517-t001:** Patient characteristics and clinical data in the unmatched and matched cohorts.

	Study Population N = 150	Control Group N = 29	Hemoadsorption Group N = 29	*p*	Post–Match Abs Std Mean Diff
		PS Matched Cohort N = 58			
Age, year	53 ± 16	55 ± 14	51 ± 15	0.291	-
Age > 70 years, *n*	17 (11.3%)	2 (6.9%)	1 (3.4%)	1.00	-
Female sex, *n*	38 (25.3%)	8 (27.6%)	6 (27.0%)	0.774	-
Body mass index, kg/m^2^	27.8 ± 5.1	28.2 ± 5.5	27.8 ± 4.6	0.717	-
Hypertension, *n*	69 (46.0%)	13 (44.8%)	10 (34.5%)	0.581	-
Coronary artery disease, *n*	60 (40.0%)	14 (48.3%)	11 (37.9%)	0.549	-
Congestive heart failure, *n*	67 (44.7%)	14 (48.3%)	16 (55.2%)	0.791	-
COPD, *n*	20 (13.3%)	3 (10.3%)	6 (20.7%)	0.453	-
Chronic liver disease, *n*	7 (4.7%)	3 (10.3%)	0 (0%)	0.250	-
Chronic kidney disease, *n* ^a^	72 (48.0%)	12 (41.4%)	14 (48.3%)	0.804	-
Diabetes mellitus, *n*	35 (23.3%)	7 (24.1%)	8 (27.6%)	1.00	-
Peripheral vascular disease, *n*	8 (5.3%)	0 (0%)	3 (10.3%)	0.250	-
Previous stroke, TIA, *n*	8 (5.3%)	2 (6.9%)	3 (10.3%)	1.00	-
Etiology of refractory cardiogenic shock					
AMI, *n*	40 (26.7%)	5 (17.2%)	3 (10.3%)	0.687	-
Acute–on–CHF, *n*	21 (14.0%)	4 (13.8%)	5 (17.2%)	1.00	-
Acute myocarditis, *n*	7 (4.7%)	0 (0%)	3 (10.3%)	0.250	-
Intoxication, *n*	3 (2.0%)	1 (3.4%)	0 (0%)	1.00	-
Severe septic shock, *n*	2 (1.3%)	1 (3.4%)	0 (0%)	1.00	-
Postcardiotomy, *n*	77 (51.3%)	18 (62.1%)	18 (62.1%)	1.00	0.142
HTx graft failure, *n*	43 (28.7%)	10 (34.5%)	12 (41.4%)	0.774	-
Cardiac arrest, *n* ^b^	21 (14.0%)	5 (17.2%)	1 (3.4%)	0.219	-
Pre–ECMO parameters					
pH	7.33 ± 0.10	7.33 ± 0.09	7.36 ± 0.09	0.439	-
Lactate, mmol/L	7.52 ± 5.35	6.90 ± 4.12	6.56 ± 4.96	0.769	-
P_(v–a)_CO_2_ gap, mmHg ^c^	8.83 ± 3.40	9.19 ± 3.03	8.47 ± 3.76	0.388	-
White blood cell, G/L	13.04 ± 7.48	11.64 ± 4.28	14.45 ± 9.57	0.146	-
C–reactive protein, mg/L	49.06 ± 67.46	31.57 ± 43.25	66.57 ± 82.23	0.054	-
Vasoactive inotropic score, point	84.9 ± 56.6	79.2 ± 51.0	90.0 ± 61.7	0.378	-
APACHE II score	30.4 ± 5.3	30.0 ± 5.5	31.1 ± 5.1	0.413	0.223
SOFA score	11.3 ± 2.3	12.2 ± 1.8	12.1 ± 2.8	0.789	0.012
SAVE score	−6.9 ± 6.1	−7.2 ± 5.6	−6.5 ± 6.7	0.668	-
VA–ECMO support					
Peripheral ECMO support, *n*	45 (30.0%)	11 (37.9%)	3 (10.3%)	0.039	-
Average ECMO flow, L/min	3.3 ± 0.5	3.5 ± 0.4	3.5 ± 0.5	0.366	0.104
ECMO support duration, hour	159 ± 67	154 ± 59	183 ± 73	0.106	-
Hemoadsorption treatment, hour	70.6 ± 8.7	0	70.5 ± 8.9	-	-

Data are presented as mean ± standard deviation and number of patients (frequency). ^a^ Chronic kidney disease was defined as estimated glomerular filtration rate < 60 mL/min/1.73 m^2^. ^b^ Cardiac arrest complicating cardiogenic shock prior to VA–ECMO implantation. ^c^ P_(v–a)_CO_2_ gap = P_v_CO_2_ − P_a_CO_2_; Normal range: 2–6 mmHg [28]. PS: propensity score; Abs Std Mean Diff: absolute standardized mean difference; COPD: chronic obstructive pulmonary disease; TIA: transient ischemic attack; AMI: acute myocardial infarction; CHF: congestive heart failure; HTx: heart transplantation; ECMO: extracorporeal membrane oxygenation; APACHE II: Acute Physiology and Chronic Health Evaluation II; SOFA: Sequential Organ Failure Assessment; SAVE: Survival after Veno–Arterial ECMO.

**Table 2 jcm-11-06517-t002:** Comparative analysis of the primary and secondary outcome parameters in the propensity score matched cohort.

Outcome Measures	Control Group N = 29	Hemoadsorption Group N = 29	*p*
	PS Matched Cohort N = 58		
Primary outcome parameters			
72-h SOFA score, point	12.1 ± 3.7	10.1 ± 3.3	0.040
In-hospital mortality, *n*	18 (62.1%)	13 (44.8%)	0.180
Secondary outcome parameters			
72-h pH	7.40 ± 0.04	7.43 ± 0.04	0.048
72-h lactate, mmol/L	2.11 ± 0.77	1.57 ± 0.96	0.015
72-h P_(v–a)_CO_2_ gap, mmHg ^a^	8.13 ± 1.26	4.47 ± 1.69	<0.001
72-h white blood cell, G/L	11.95 ± 4.32	11.35 ± 6.16	0.650
72-h C-reactive protein, mg/L	140.05 ± 86.72	116.69 ± 55.33	0.159
72-h vasoactive inotropic score, point	35.2 ± 36.1	13.8 ± 19.5	0.007
Bleeding/72 h, *n* ^b^	22 (75.9%)	13 (44.8%)	0.049
Reoperation for bleeding/72 h, *n*	9 (31.0%)	7 (24.1%)	0.754
PRC transfusion/72 h, unit	9 ± 9	10 ± 6	0.461
AKI_total_ within 72 h, *n*	21 (72.4%)	21 (72.4%)	1.00
Renal replacement therapy within 72 h, *n*	15 (51.7%)	19 (65.5%)	0.481
Mechanical ventilation, day	30.3 ± 39.2	34.6 ± 30.3	0.673
Length of ICU stay, day	37 ± 45	37 ± 23	0.962
Length of Hospital stay, day	49 ± 59	45 ± 33	0.707

Data are presented as mean ± standard deviation and number of patients (frequency). ^a^ P_(v–a)_CO_2_ gap = P_v_CO_2_ − P_a_CO_2_; Normal range: 2–6 mmHg [28]. ^b^ Clinically relevant blood loss required conservative (i.e., blood products and factor concentrates) and/or surgical therapy (registered for the post–VA–ECMO implantation period). PS: propensity score; SOFA: Sequential Organ Failure Assessment; PRC: packed red cell; AKI: acute kidney injury; ICU: intensive care unit.

## Data Availability

The data underlying this article will be shared on reasonable request to the corresponding author.

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
