# Peer review of "Influence of Venoarterial Extracorporeal Membrane Oxygenation Integrated Hemoadsorption on the Early Reversal of Multiorgan and Microcirculatory Dysfunction and Outcome of Refractory Cardiogenic Shock"

_jcm, 2022, doi:10.3390/jcm11216517_

Round 1

Reviewer 1 Report

I have read with attention the paper by Adam Soltesz and coauthors. The paper revealed important informations related the impact of hemoadsorption on microcirculatory dysfunction in patients with CS supported with VA ECMO. 

Some additional informations are required: 

1) More informations regarding pharmacological circulatory support should be included please add in table 1 an inotropic score to measure the median vasopressor load in the two group. Furthermore include informations how it is changed at 72 hours. 

2)How many patients had a cardiac arrest in control group and in the treatment group please add these data to Table 1 

3)Had some patients need of CKRT? Please include this information for both group. 

Reviewer 2 Report

     Thank you for the opportunity to review this article. Congratulation for this work. The important finding is this investigation is the lower mortality rate and the rapid stabilization of hemodynamic, inflammatory, and clinical condition in refractory CS patients who received hemoadsportion therapy. The study cohort is limited but the findings are encouraging especially for this cohort with usually high mortality rate in the literatur.

-        The title is long and need to be shortened.

-        The authors stated that hemoadsorption therapy was applied for an overall 72 hours. How often was the adsorber changed? What was the decision to interrupt the hemoadsorption therapy after 72 hours?

-        Are there any data available regarding the time of occurrence of refractory CS in both groups after initial therapy?

-        Any other inflammatory markers measured at your institution (e.g. IL6 etc…..)?

-        How many patients received peripheral or central cannulation in both groups? The later is more invasive and may impact the course.

-        How do you explain that bleeding complications were significantly reduced in the hemoadsorption group. This is an interesting finding and should be more highlighted. Is this an indirect effect of controlling the inflammatory situation and controlling inflammation related to the ECMO circuit and therefore gaining more control in terms of diffuse bleeding at surgical sites?

-        Although hemodynamic and clinical stabilization were significantly more rapid in the hemoadsorption group, ICU stay and ECMO duration were similar in both groups. It is expected to have a shorter ECMO duration and consequently a shorter ICU stay when hemodynamic stability occurs. Is this related to your ECMO Weaning protocol? It may be helpful to include your weaning protocol in such patients.

-         Pre- and post-treatment Data regarding changes in catecholamine dosage in both groups are missing. Please add this information.

This submission has potential but there are questions/concerns that need to be addressed. I look forward to reviewing the author's responses and revised manuscript.

Round 2

Reviewer 2 Report

The authors responded to all queries accordingly. Congratulations for this great work.